# The Molecular Composition of Peptide Toxins in the Venom of Spider *Lycosa coelestis* as Revealed by cDNA Library and Transcriptomic Sequencing

**DOI:** 10.3390/toxins15020143

**Published:** 2023-02-10

**Authors:** Xiangyue Wu, Yan Chen, Hao Liu, Xiangjin Kong, Xinyao Liang, Yu Zhang, Cheng Tang, Zhonghua Liu

**Affiliations:** 1The National and Local Joint Engineering Laboratory of Animal Peptide Drug Development, College of Life Sciences, Hunan Normal University, Changsha 410081, China; 2Department of Philosophy, Peking University, Beijing 100871, China

**Keywords:** *Lycosa coelestis*, cDNA library, peptide toxin, diversity, transcriptome sequencing

## Abstract

In the so-called “struggle for existence” competition, the venomous animals developed a smart and effective strategy, envenomation, for predation and defense. Biochemical analysis revealed that animal venoms are chemical pools of proteinase, peptide toxins, and small organic molecules with various biological activities. Of them, peptide toxins are of great molecular diversity and possess the capacity to modulate the activity of ion channels, the second largest group of drug targets expressed on the cell membrane, which makes them a rich resource for developing peptide drug pioneers. The spider *Lycosa coelestis* (*L. coelestis*) commonly found in farmland in China is a dominant natural enemy of agricultural pests; however, its venom composition and activity were never explored. Herein, we conducted cDNA library and transcriptomic sequencing of the venom gland of *L. coelestis*, which identified 1131 high-quality expressed sequence tags (ESTs), grouped into three categories denoted as toxin-like ESTs (597, 52.79%), cellular component ESTs (357, 31.56%), and non-matched ESTs (177, 15.65%). These toxin-like ESTs encode 98 non-reductant toxins, which are artificially divided into 11 families based on their sequence homology and cysteine frameworks (2–14 cysteines forming 1–7 disulfide bonds to stabilize the toxin structure). Furthermore, RP-HPLC purification combined with off-line MALDI-TOF analysis have detected 147 different peptides physically existing in the venom of *L. coelestis*. Electrophysiology analysis confirmed that the venom preferably inhibits the voltage-gated calcium channels in rat dorsal root ganglion neurons. Altogether, the present study has added a great lot of new members to the spider toxin superfamily and built the foundation for characterizing novel active peptides in the *L. coelestis* venom.

## 1. Introduction

During hundreds of millions of years of evolution, many animal species have evolved venom glands to produce venom, a complex of proteinase, peptide toxins, and small organic molecules, for their predation and defense [1]. It is estimated that at least 220,000 venomous animal species live on the earth [2], and most of them are spiders, jellyfish, centipedes, scorpions, snakes, and cone snails. Spiders are the most representative toxic arthropods, and their nearly perfect evolution makes them well-adapted to different kinds of environments, including rainforests, deserts, grassland, and so on [3]. Nowadays, more than 50,751 spider species have been reported (World Spider Catalog Version 24). The peptide toxins with molecular weight (MW) less than 10 kDa in the spider venoms are of great molecular diversity [3,4,5], and they are rich in cysteines which formed intra-molecular disulfide bonds to stabilize their globular structures [6]. This cysteine-rich structure property was used as an important criterion to identify toxins derived from venom gland cDNAs. Meanwhile, the cysteine frameworks in spider toxins are relatively conserved despite great amino acid variation in the non-cysteine positions in the peptide sequence; this might be because the disulfide bonds confer spider toxins the ultra-stability resistant to acid, alkaline, heat, and proteinase digestion. There are three commonly found disulfide bond modes in spider toxins [7,8,9]: (i) the inhibitor cysteine knot (ICK) motif with the disulfide bond connection of C_1_-C_4_, C_2_-C_5_, and C_3_-C_6_ (C represents cysteine, and the subscripted number indicates the relative position of cysteines in the sequence), in which the C_3_-C_6_ disulfide bond crosses the loop formed by C_1_-C_4_, C_2_-C_5,_ and their connecting peptide; (ii) the disulfide directed hairpin (DDH) motif with the disulfide bonds of C_1_-C_3_, C_2_-C_5_, and C_4_-C_6_; and (iii) the Kunitz motif with the disulfide bonds of C_1_-C_6_, C_2_-C_4_, and C_3_-C_5_. Based on the great number of spider species on earth and a conservative estimate of 200 different peptides in each species’ venom [10], the summed number of spider toxins is astronomical. However, approximately only 0.01% of spider toxins have been characterized to date [11]. The collection of enough spider venom for the subsequent purification and activity assay is always challenging as: (i) most spider species are of small body size and only a small amount of venom is milked by each animal; and (ii) limited spiders can be captured in the wild. Alternatively, the cDNA library sequencing of spider venom gland can efficiently uncover the sequence information of peptide toxins in the venom, with the cost of just several animals. These sequence data derived from the toxin encoding cDNAs could be used in the following two aspects: (i) with the development of protein structure predicting tools [12], it is becoming practical to virtually screen peptide modulators for a given target by the computing strategy based on the peptide toxin sequence database; (ii) some smart strategies, such as phage-displayed and tethered-toxins screening, have been validated to be powerful in identifying modulators for a given target from the peptide toxin sequence database without getting their purified protein molecule entities [13,14]. 

Spiders are one of the most successful arthropod predators. Their venom has been proven to be a rich source of hyperstable insecticidal mini-proteins that cause insect paralysis or lethality through the modulation of ion channels, receptors, and enzymes [15]. Interestingly, these venom peptide toxins are also proved as high-affinity and high-specificity modulators of many mammalian protein targets, such as G-protein coupled receptors [16,17], ion channels [18], and proteinase [19,20], which consequently provides a rich mine for developing drug pioneers acting on these disease therapeutical targets and identifying molecular probes for studying their structure-function relationships [21,22,23]. The traditional strategy for identifying active peptides in the venom involves an activity-guided purification procedure followed by sequencing by Edman degradation, which is always costly and hard to get the full sequence. Fortunately, this problem can be easily resolved by the combined use of Edman degradation to get the partial N-terminal sequence, cDNA library searching, and the precise MW analysis. Meanwhile, this hybrid strategy is also powerful in uncovering the sequences of peptide toxin entities in the venom in peptidomics studies [24]. Therefore, the venom gland cDNA library sequencing is of great significance for accelerating the discovery of venom-derived active peptides. 

The spider *Lycosa coelestis* (*L. coelestis*), commonly found in many provinces in China (Taiwan, Sichuan, Zhejiang, Guizhou, Hebei, etc.), belongs to the *Lycosidae* family and is a dominant natural enemy for farmland pests. However, the peptide toxin composition in the venom of *L. coelestis* has never been explored. Herein, we investigated the molecular composition of peptide toxins in the venom of *L. coelestis* by venom gland cDNA library and transcriptomic sequencing, which identified 98 toxin-like peptides grouped into 11 families. Meanwhile, RP-HPLC purification combined with MALDI-TOF MS analysis has detected 147 different peptide toxin entities physically existing in the *L. coelestis* venom. Electrophysiology studies showed that *L. coelestis* venom preferably inhibits the voltage-gated calcium channels (CaVs) in rat dorsal root ganglion (DRG) neurons. Taken in all, the present study has added a great number of novel peptide toxins to the spider toxin superfamily and built the foundation for screening novel active peptides in the *L. coelestis* venom. 

## 2. Results and Discussion 

### 2.1. General Features and Annotation of L. coelestis Venom Gland ESTs

Clone sequencing of *L. coelestis* (Figure 1A) venom gland cDNA library resulted in 1088 high-quality expressed sequence tags (ESTs), which were grouped into three categories by blasting against the non-reductant protein database (https://blast.ncbi.nlm.nih.gov/Blast.cgi?PROGRAM=blastx&PAGE_TYPE=BlastSearch&LINK_LOC=blasthome; Accessed on 20 November 2022): 554 toxin-like ESTs (cDNA length ranging from 0.43–0.90 Kb), 357 cellular component ESTs (cDNA length ranging from 0.35–0.97 Kb), and 177 non-matched ESTs (cDNA length ranging from 0.36–0.95 Kb) that cannot be linked to other proteins in the database. With a special interest in the cDNAs encoding peptide toxins in the venom gland, we extracted 43 more toxin-like ESTs from the transcriptomic sequencing data, which were pooled with the cDNA library-derived ESTs for the following analysis. Interestingly, significantly fewer toxin-like ESTs were identified by the transcriptomic analysis, suggesting the more purposeful cDNA library sequencing method is more powerful in discovering the toxin-like proteins. These two strategies are complementary in discovering peptide toxin precursors. Altogether, function annotation confirmed that the toxin-like ESTs were the most abundant in the database (50.92%), and cellular component ESTs and non-matched ESTs accounted for less (32.81% and 16.27%, respectively) (Figure 1B). 

### 2.2. Cluster Analysis of L. coelestis Venom Gland ESTs 

The ESTs’ abundance in the cDNA library partially reflects the expression level of their encoding proteins, in particular, peptide toxins are highly expressed in the venom gland and their mRNAs are always of the high copy. Therefore, we conducted a cluster analysis of the cDNA library sequencing-derived ESTs (Figure 2A). These 1088 ESTs were clustered into 400 clusters, including 298 singletons and 102 contigs of different artificially defined sizes (2–5, 6–10, 11–15, 16–30, and >30). Firstly, the 554 toxin-like ESTs were grouped into 26 clusters comprising 10 singletons and 16 contigs: (1) 8 contigs of size 2–5 contain 28 ESTs, which represent 19 unique genes encoding 15 proteins; (2) 4 contigs of size 6–10 contain 32 ESTs, which represent 14 unique genes encoding 12 proteins; (3) 1 contig of size 11–15 contain 14 ESTs, which represent 4 unique genes encoding 2 proteins; (4) 1 contig of size 16–30 contain 17 ESTs, which represent 3 unique genes encoding 1 protein; (5) 2 contigs of size >30 contain 453 ESTs, which represent 56 unique genes encoding 38 proteins. The identities of the representative peptide toxins in the toxin-like ESTs’ clusters were shown in Figure 2B, which indicates that LcTx-17-like toxins in family F and LcTx-5-like toxins in family E (see family analysis below) are the most abundant in the *L. coelestis* venom gland. 

Secondly, the 357 cellular component ESTs were grouped into 219 clusters, including 150 singletons and 69 contigs: (1) 65 contigs of size 2–5 contain 179 ESTs, which represent 117 unique genes encoding 102 proteins; and (2) 4 contigs of size 6–10 contain 28 ESTs, which represent 11 unique genes encoding 6 proteins. Contigs of size ≥11 were not found in this category (Figure 2A). 

Thirdly, the 177 non-matched ESTs were grouped into 155 contigs, including 138 singletons and 17 contigs of size 2–5 containing 39 ESTs, which represent 27 unique genes encoding 27 proteins. No contigs of size ≥6 were presented in this category (Figure 2A). 

Consequently, most toxin-like ESTs (98.19%) were clustered into contigs, in contrast to the lower clustering proportion observed in the cellular component ESTs (57.98%) and the non-matched ESTs (22.03%). This is in accordance with the concept that most peptide toxin transcripts are of high copy in the venom gland. Taken together, a total of 521 putative non-reductant proteins were identified in the *L. coelestis* venom gland, including 98 toxin-like proteins (including 21 toxin-like proteins derived from transcriptomic sequencing), 258 cellular component proteins, and 165 non-matched proteins without significant homology to proteins in the database. 

### 2.3. Family Analysis of Putative Toxin Precursors in L. coelestis Venom Gland

The *L. coelestis* venom gland cDNA-derived toxin-like peptides were named LcTx-n, in which Lc, Tx, and n represent *L. coelestis*, toxin, and the clone number, respectively. The capital letter ‘P’ and ‘T’ is added to the end of the toxin name to indicate that it is a partial sequence (LcTx-n-P) or derived from transcriptomic sequencing (LcTx-n-T), respectively. Sequence analysis confirmed that all cDNA library-derived ESTs contain the poly-A tail, suggesting their encoding toxin precursors have the complete C-termini (i.e., their mature peptides have the complete C-termini). All the 98 LcTxs were grouped into 11 families based on their sequence homology and cysteine frameworks (Figure 3). The nucleotide sequences of these toxin-like peptides were deposited in the Genbank database (https://www.ncbi.nlm.nih.gov/genbank/; Genbank accession numbers: OQ207192—OQ207289). 

#### 2.3.1. Family A

Family A contains 3 toxin precursors, LcTx-64, LcTx-965, and LcTx-1436-P-T. Among them, LcTx-64 and LcTx-965 have a typical signal peptide but no propeptide, they differ by only one amino acid in their mature peptides, and are best aligned with two toxins with unknown biological functions, U31-ctenitoxin-Cs1c (77–78% identity) from the spider *Cupiennius salei* and U9-ctenitoxin-Pr1a (71–73% identity) from the spider *Phoneutria reidyi*, in the NR (Non-Redundant) and UniProt database, respectively. LcTx-1436-P-T is derived from transcriptomic sequencing and has a truncated N-terminus, it is best aligned with U30-ctenitoxin-Cs1a (60% identity, NR database) from the spider *Cupiennius salei* and U9-ctenitoxin-Pr1a (61% identity, UniProt database) from the spider *Aaneus ventricosus*, both of which are functionally undermined. All members in this family share a conserved cysteine framework, C_1_-C_2_-C_3_C_4_-C_5_-C_6_-C_7_-C_8_-C_9_-C_10_, which is also found in the U31-ctenitoxin-Cs1c [25]. Therefore, their disulfide mode is predicted to be C_1_-C_4_, C_2_-C_5_, C_3_-C_7_, C_6_-C_9_, and C_8_-C_10_ by similarity. 

#### 2.3.2. Family B

Family B contains 4 toxin precursors which are divided into two subgroups, with LcTx-799 and LcTx-693 being best aligned with U20-lycotoxin-Ls1c (88–89% identity), and LcTx-1421-T and LcTx-1432-T best aligned with U14-lycotoxin-Ls1b (80–86% identity) from the spider *Lycosa singoriensis,* respectively. These toxins are made of signal peptides and mature peptides without the propeptide. Except for LcTx-799, which has its second cysteine mutated to arginine, they all share a conserved cysteine framework of C_1_-C_2_-C_3_C_4_-C_5_-C_6_C_7_-C_8_-C_9_-C_10_. The ‘hit’ toxins U20-lycotoxin-Ls1c and U14-lycotoxin-Ls1b are predicted to have anti-bacterial activity; therefore, toxins in this family are likely antimicrobial peptides [26]. It is unknown whether LcTx-1421-T and LcTx-1432-T have a complete C-terminus. 

#### 2.3.3. Family C

Family C contains 5 toxin precursors and only LcTx-1031 is derived from cDNA library sequencing. LcTx-1031 and LcTx-1411-T are best blasted to putative neurotoxin LTDF-S-04 (78–79% identity, NR database) from the spider *Dolomedes fimbriatus* but have no ‘hit’ toxins in the UniProt database. These two toxins have the cysteine framework of C_1_-C_2_-C_3_-C_4_-C_5_-C_6_C_7_-C_8_-C_9_-C_10_-C_11_-C_12_. It is unknown whether LcTx-1411-T has the complete C-terminus. On the other hand, LcTx-1427-T, LcTx-1426-T, and LcTx-1408-P-T are best aligned with U24-ctenitoxin-Pn1a like protein (81% identity) from the spider *Argiope bruennichi*, putative neurotoxin LTDF-S-18 (78% identity) from the spider *Dolomedes* fimbriatus, and U24-ctenitoxin-Pn1a (65% identity) from the spider *Araneus ventricosus,* in the NR database, respectively. These ‘hit’ toxins’ biological functions are undetermined. LcTx-1427-T has the cysteine framework of C_1_-C_2_-C_3_-C_4_-C_5_-C_6_-C_7_-C_8_-C_9_-C_10_. Being different from LcTx-1031 and LcTx-1411-T, the toxins LcTx-1426-T and LcTx-1408-P-T possess another cysteine framework, C_1_-C_2_-C_3_-C_4_-C_5_-C_6_-C_7_-C_8_-C_9_-C_10_-C_11_-C_12_. 

#### 2.3.4. Family D

Family D contains 4 toxin precursors, all of which have the typical propeptide. LcTx-107 shows relatively high sequence homology (75% identity, NR database) with putative neurotoxin LTDF-S-05 from the spider *Dolomedes fimbriatus* and has the classical cysteine framework commonly found in ICK motif toxins (C_1_-C_2_-C_3_C_4_-C_5_-C_6_). The other three highly homologous toxins in this family are LcTx-67, LcTx-1001, and LcTx-670, which are best aligned with Omega-lycotoxin-Gsp2671a, which was identified as a modulator of the CaV2.1 P-type voltage-gated calcium channel [27]. These three toxins share a conserved cysteine framework of C_1_-C_2_-C_3_C_4_-C_5_-C_6_-C_7_-C_8_. All toxins in this family should adopt an ICK motif folding (C_1_-C_4_, C_2_-C_5_, and C_3_-C_6_ in LcTx-107; C_1_-C_4_, C_2_-C_5_, C_3_-C_8_, and C_6_-C_7_ in the other three), and the fourth pair of disulfide bond (C_6_-C_7_) further stabilizes the structure [28]. 

#### 2.3.5. Family E

Family E is the most abundant cluster in the library which contains 32 precursor toxins. Except for LcTx-1418-P-T which is an N-terminus truncated peptide derived from transcriptomic sequencing, all the other toxins are derived from cDNA library sequencing and are of full length. A classical amidation signal, the most C-terminus glycine residue, is presented in their mature peptides, indicating the mature peptides are amidated. Moreover, although LcTx-1418-P-T and LcTx-891 share high sequence homology with the other members, their sequence mutations in the processing quadruplet motif [29,30] (PQM) make them lack a typical propeptide cutting site. All toxins in this family, except LcTx-1032 with its 7th cysteine mutated to phenylalanine, have the conserved cysteine framework of C_1_-C_2_-C_3_C_4_-C_5_-C_6_-C_7_-C_8_ in their mature peptides, which should adopt an ICK motif folding as well (C_1_-C_4_, C_2_-C_5_, C_3_-C_8_, and C_6_-C_7_). These toxins are further divided into two subgroups, with LcTx-897, LcTx-1084, LcTx-202, and LcTx-164 being best aligned with U1-lycotoxin-Ls1b (81–85% identity), from the spider *Lycosa singoriensis.* The others matched with U2-lycotoxin-Lt19b, from the spider *Lycosa tarantula*. We were unable to predict the function of these toxins due to the unknown biological activities of the ‘hit’ toxins in the database. Moreover, it is likely these toxins should play an important role in the predation and defense of *L. coelestis* as EST cluster analysis showed that their transcripts account for 64.26% (356 out of 554) of the total toxin-like ESTs in cDNA library sequencing (Figure 2B, the LcTx-5-like toxins). 

#### 2.3.6. Family F

Family F contains 10 toxin precursors, all of them have the typical propeptide and their mature peptides share a conserved cysteine framework of C_1_-C_2_-C_3_C_4_-C_5_-C_6_-C_7_-C_8_. As those toxins in Families D and E, their proposed disulfide model is also C_1_-C_4_, C_2_-C_5_, C_3_-C_8_, and C_6_-C_7_. Moreover, the most C-terminus glycine residue presented in Family F toxins (except LcTx-731) indicates that their mature peptides are amidated. Blasting analysis showed that LcTx-1216, LcTx-580, LcTx-1022, LcTx-17, LcTx-683, LcTx-62, and LcTx-1229 possess extremely high sequence homology with the toxin U4-lycotoxin-Ls1a from the spider *Lycosa singoriensis* (92–93% identity). This ‘hit’ toxin was proposed to enhance the high-affinity desensitization of human P2RX3 purinoceptors [26], indicating the possible same function for this subgroup of toxins. As the LcTx-5-like toxins in family E, the LcTx-17-like toxins in this family are also of high abundance in the venom gland as revealed by analyzing their transcripts’ abundance (accounting for 17.51% of the toxin-like ESTs in cDNA library sequencing). LcTx-321 is best aligned with U3-lycotoxin-Ls1a (76% identity) from the spider *Lycosa singoriensis*. LcTx-103 in the second subgroup is best aligned with toxin-like structure LSTX-D6 (83% identity) from the spider *Lycosa singoriensis*. LcTx-731 in the third subgroup is best matched with U3-lycotoxin-Ls1u from the spider *Lycosa singoriensis* as well (81% identity). 

#### 2.3.7. Family G

Family G contains 14 toxin precursors with the same cysteine framework of C_1_-C_2_-C_3_C_4_-C_5_-C_6_-C_7_-C_8_, and the mature toxins should also adopt an ICK motif folding. This family of toxins are relatively diverse and could be divided into 4 subgroups: (1) LcTx-389, LcTx-435, and LcTx-320 do not have typical propeptide and their mature peptides are highly homologous which differ from each other by a single-site mutation; these toxins are best aligned with U2-lycotoxin-Lt19c (83–84% identity, NR database) from the spider *Lycosa tarantula*; (2) LcTx-1127-P and LcTx-384 are derived from cDNA library sequencing; however, LcTx-1127-P seems to be a partial sequence with its N-terminus truncated but with the complete C-terminus. These two toxins do not have typical propeptide and have moderate sequence homology with U7-lycotoxin-Ls1c (51–55% identity, NR database) from the spider *Lycosa singoriensis*; (3) LcTx-927, LcTx-1419-T, LcTx-460, LcTx-499, and LcTx-1403-T are best aligned with U8-lycotoxin-Ls1s (70–75% identity) from the spider *Lycosa singoriensis*, and all of them have the typical propeptide; and 4) LcTx-899, and LcTx-1435-T are best aligned with U6-lycotoxin-Ls1g from the spider *Lycosa singoriensis* (61–63% identity). LcTx-1415-P-T and LcTx-1401-T are best aligned with U6-lycotoxin-Ls1d from the spider *Lycosa singoriensis* (58–71% identity). LcTx-1415-P-T is N-terminus truncated lacking the signal peptide and propeptide as compared to LcTx-1435-T. 

#### 2.3.8. Family H

Family H contains 10 toxin precursors which have diverse cysteine frameworks: (1) LcTx-584, LcTx-281, and LcTx-147 in this family but not the others do not have a classical propeptide in their sequences. They share a cysteine framework of C_1_-C_2_-C_3_C_4_-C_5_-C_6_-C_7_-C_8-_C_9_, and they are best aligned with putative neurotoxin LTDF-S-11 from the spider *Dolomedes fimbriatus* (80% identity); (2) LcTx-1424-T, LcTx-581, and LcTx-839 share a cysteine framework of C_1_-C_2_-C_3_-C_4_C_5_-C_6_-C_7_-C_8_-C_9_-C_10_. These toxins are best aligned with U12-lycotoxin-Ls1a (64–85% identity, NR database) from the spider *Lycosa singoriensis*. LcTx-581 and LcTx-839 differ from each other in the propeptide region; and (3) LcTx-1402-T, LcTx-1130, LcTx-574, and LcTx-1178 have the same cysteine framework of C_1_-C_2_-C_3_-C_4_C_5_-C_6_-C_7_-C_8_-C_9_-C_10_-C_11_-C_12_; their mature peptides should be amidated as indicated by the amidation signal at their C-termini (the most C-terminus glycine residue). LcTx-1402-T and LcTx1130 are best aligned with U13-lycotoxin-Ls1c from the spider *Lycosa singoriensis* (76–77% identity, NR database), LcTx-574 is best matched with Tx-37, U23-ctenitoxin-Cs1 from the spider *Cupiennius salei* (64% identity, NR database), while LcTx-1178 shows the highest sequence homology with putative neurotoxin LTDF-S-14 from the spider *Dolomedes fimbriatus* (79% identity, NR database). 

#### 2.3.9. Family I

Family I contains 3 precursor toxins; they all have typical propeptides and their mature peptides should also be amidated as indicated by the C-terminus glycine amidation signal. This family of toxins have the most cysteines with the cysteine framework of C_1_-C_2_-C_3_-C_4_C_5_-C_6_-C_7_-C_8_-C_9_-C_10_-C_11_-C_12_-C_13_-C_14_. Among them, LcTx-925 and LcTx-715 differ from each other by only one amino acid mutation in their mature peptides (arginine ↔ glutamine mutation), and LcTx-51 has a shorter signal peptide but the same mature peptide as LcTx-715. These toxins are best aligned with Omega-CNTX-Pn3a from the spider *Phoneutria nigriventer* (43% identity, Uniprot database). The Omega-CNTX-Pn3a toxin is identified as an irreversible antagonist of the CaV2.1/CaV2.2 channels, while it reversibly inhibits the CaV2.3 channel and spares the CaV3 channels [31]. Based on the cysteine framework similarity with Omega-CNTX-Pn3a, the disulfide model for toxins in this family should be: C_1_-C_5_, C_2_-C_6_, C_3_-C_10_, C_4_-C_9_, C_7_-C_8_, C_11_-C_12_, and C_13_-C_14_. 

#### 2.3.10. Family J

This family contains 5 toxin precursors, all of which have the propeptides and the C-terminus amidation signals (C-terminus -G, -GK, or -GGK motif); therefore, their mature peptides are supposed to be amidated. These toxins were divided into two subgroups based on their sequence homology and cysteine frameworks: (1) LcTx-85 and LcTx-45 are best aligned with U2-lycotoxin-Ls1c (82–83% identity, NR database) from the spider *Lycosa singoriensis*, as these two toxins only have one amino acid difference in their signal peptides. The U2-lycotoxin-Ls1c toxin is insecticidal to house crickets by inducing an excitatory slow-onset but irreversible spastic paralysis; however, it is not toxic to mice when intracranially injected (at a dose of 0.5 ug/g) [26]. This toxin also inhibits the human voltage-gated potassium channel Kv1.5 most likely by acting as a gating modifier. LcTx-85 and LcTx-45 have the same cysteine framework (C_1_-C_2_-C_3_-C_4_-C_5_-C_6_-C_7_-C_8_), and possibly the same disulfide model as U2-lycotoxin-Ls1c (C_1_-C_4_, C_2_-C_8_, C_3_-C_7_, and C_5_-C_6_), based on their high sequence homology; (2) LcTx-283 and LcTx-330 have only one amino acid difference in their mature peptides, and LcTx-1186, LcTx-283, and LcTx-330 are best matched with U16-lycotoxin-Ls1a from the spider *Lycosa singoriensis* (53–56% identity, NR database), which was proposed to have ion channel inhibiting activity but not experimentally validated yet. The cysteine framework of these three toxins was as that observed in families D-F (C_1_-C_2_-C_3_C_4_-C_5_-C_6_-C_7_-C_8_)_,_ for which the proposed disulfide mode is C_1_-C_4_, C_2_-C_5_, C_3_-C_8_, and C_6_-C_7_. 

#### 2.3.11. Family K

Family K contains 8 diverse toxins with very low sequence homology to each other. Only LcTx-681 and LcTx-129-P, but not the others, are derived from cDNA library sequencing, and signal peptides are only recognized in the sequences of LcTx-1433-T, LcTx-1406-P-T, and LcTx-681; no propeptide was presented in their sequences. LcTx-129-P only has two cysteines (C_1_-C_2_) and is best aligned with hypothetical protein X975_03585, partial (68% identity, NR database) from the spider *Stegodyphus mimosarum*. LcTx-681 has a rather long mature peptide that contains four cysteines (C_1_-C_2_-C_3_-C_4_), and it matches the best with u21-ctenitoxin-Pn1a (43% identity, NR database) from the spider *Trichonephila clavipes*. LcTx-1433-T and LcTx-1430-P-T have the same cysteine framework (C_1_-C_2_-C_3_-C_4_-C_5_-C_6_C_7_-C_8_), and they have the highest sequence homology with putative neurotoxin LTDF 15-02 (56% identity, NR database) from the spider *Dolomedes fimbriatus* and Tx-1159-T (46% identity, NR database) from the spider *Heteropoda pingtungesis*, respectively. LcTx-1406-P-T, LcTx-1429-P-T, and LcTx-1407-P-T have ten cysteines in their sequences, but the cysteine frameworks are slightly different (C_1_-C_2_-C_3_C_4_-C_5_-C_6_-C_7_-C_8_-C_9_-C_10_ for LcTx-1429-P-T and LcTx-1407-P-T, and C_1_-C_2_-C_3_C_4_-C_5_-C_6_C_7_-C_8_-C_9_-C_10_ for LcTx-1406-P-T). Blast analysis showed that LcTx-1406-P-T is best aligned with hypothetical protein NPIL_628861 from the spider *Nephila pilipes* (44% identity), LcTx-1429-P-T aligned with U2-ctenitoxin-Vf2 from the spider *Viridasium fasciatus* (75% identity), and LcTx-1407-P-T aligned with U3-aranetoxin-Ce1a from the spider *Parasteatoda tepidariorum* (45% identity), respectively. LcTx-1412-P-T has a cysteine framework of C_1_-C_2_-C_3_C_4_-C_5_-C_6_-C_7_-C_8_ and is supposed to be an ICK motif toxin, as it matches the best with U8-agatoxin-Ao1a (81% identity) from the spider *Parasteatoda tepidariorum*. 

### 2.4. RP-HPLC Purification of L. coelestis Venom and Off-Line MALDI-TOF Mass Spectrometry Analysis of the Eluted Fractions

It is hard to purify each of the peptide toxin in the venom to homogeneity even by combining multiple purification strategies (size exclusion chromatography, RP-HPLC, ion-exchange chromatography, and so on). Fortunately, the peptide mass fingerprint as determined by MALDI-TOF MS analysis was commonly used for revealing the diversity of venom peptidomes and could be a potential strategy for chemotaxonomy, with the count of the molecular species presented in the venom as revealed by MALDI-TOF analysis being an indicator of the molecular diversity of venom peptides [32,33,34]. To uncover the molecular diversity of peptide toxins presented in the *L. coelestis* venom, we performed RP-HPLC purification of the venom followed by MALDI-TOF analysis of each eluted fraction. As shown in Figure 4A, 51 fractions were collected and most of them are with a retention time of 21–50 min (acetonitrile gradient from 26% to 55%). MALDI-TOF analysis has detected 147 different molecular weights (Figure 4B), suggesting a great number of peptide toxins physically existing in the venom. 

### 2.5. Activity of L. coelestis Venom against the Voltage-Gated Na^+^, Ca^2+^ and K^+^ Channels in DRG Neurons

Spider venoms are rich in peptide toxins with potential modulatory activities on various ion channels. We have tested the activities of several spider venoms on endogenous ion channels expressed in primary cells in our previous studies [35,36,37,38]. To gain a glimpse into the activity of *L. coelestis* venom on ion channels, we analyzed its effect on the whole-cell currents of the endogenous voltage-gated K^+^, Na^+^, and Ca^2+^ (K_V_, NaV, and CaV) channels in dorsal root ganglion (DRG) neurons, which are gates for nociceptive signal transmission from peripheral to the central nervous system [39]. As shown in Figure 4C,D, *L. coelestis* venom at a concentration of 1 μg/μL inhibited the currents of K_V_, NaV, and CaV channels by 7.45 ± 3.16%, 18.54 ± 3.23%, and 32.39 ± 3.12%, respectively. As the venom preferably inhibits the CaV channels, which are mainly contributed by the N-type CaV2.2 and T-type CaV3.1–3.3 channels involved in pain [40,41], these data suggest that it is promising to characterize novel analgesic peptides from the *L. coelestis* venom. 

## 3. Conclusions

Spider venoms are rich in various kinds of peptide toxins with diverse bioactivities, of which their high-affinity and high-specificity modulation of the function of ion channels make them valuable for developing drug pioneers in treating channelopathies [42]. The present study has uncovered the molecular composition of the peptide toxins in the venom of the spider *L. coelestis* by venom gland cDNA library and transcriptomic sequencing, which built the foundation for further discovering the active peptides in the venom. Moreover, preliminary electrophysiology experiments showed that the venom preferably inhibits the CaV channels in DRG neurons, suggesting it is promising to isolate the active peptidic CaV channel antagonists from the venom, which might be used for pain treatment. 

## 4. Materials and Methods

### 4.1. cDNA Library and Transcriptomic Sequencing of the Venom Gland from the Spider L. coelestis

The spiders *L. coelestis* were captured in Guangxi province in China and shortly housed in our lab; venom milking was performed by an electrical stimulation method and approximately 500 mg crude venom was collected from 800 spiders. Four days post milking, venom glands from 7 spiders were dissected and immediately homogenized in liquid nitrogen. Total RNA was extracted, and the cDNA library was constructed using the SMART^®^ cDNA library Construction kit following the manufacturer’s instructions (Takara Bio USA, Inc., Mountain View, CA, USA). The primary cDNA library was diluted by 10^6^ folds and inoculated onto LB agar plates with ampicillin (100 μg/mL), incubated at 37 °C overnight. Bacteria clones were randomly picked and the inserted sequence in the library vector between the designated Sfi I sites was sequenced using M13F forward universal sequencing primer. Inserted sequences of >300 bp length were defined as high-quality expressed sequence tag (EST). We performed rounds of clone sequencing until no new EST was identified. Transcriptomic sequencing was performed in Illumina HiSeq X Ten platform (Illumina, San Diego, CA, USA) in Oebiotech (Shanghai OE Biotech. Co., Ltd., Shanghai, China). Briefly, total venom gland RNA (approximately 6 µg; A260/280 = 2.15, A260/230 = 1.79, RIN (RNA integrity number) is 9.4) was extracted, and the library was constructed using the TruSeq Stranded mRNA LTSample Prep Kit (Illumina, San Diego, CA, USA) following the manufacturer’s instructions. After removing the adapters and low-quality reads, short reads were assembled using Trinity [43] (version: 2.4) and the longest transcript was selected as the unigene, based on similarity and length analysis. Finally, the coding sequence and protein sequence database were constructed by BLAST and ESTscan analysis, and transcripts encoding the toxin-like peptides were extracted and mixed with ESTs derived from cDNA library sequencing for the subsequent bioinformatic analysis. 

### 4.2. EST Translation and Annotation

ESTs were blasted against the non-reductant (NR) protein database using the NCBI blast tool (https://blast.ncbi.nlm.nih.gov/Blast.cgi?PROGRAM=blastx&PAGE_TYPE=BlastSearch&LINK_LOC=blasthome; accessed on 20 November 2022) and classified into three categories based on function annotations of their ‘hit’ sequences in the database: (1) toxin-like peptide ESTs; (2) cellular component ESTs; and (3) non-matched ESTs. The corresponding toxin-like protein database was constructed by translating the ESTs using the translation tool (https://web.expasy.org/translate/; accessed on 22 November 2022), and reductant protein sequences were removed. 

### 4.3. Cluster Analysis of ESTs and Family Classification of the Toxin-Like Peptides

ESTs from each group (toxin-like, cellular component, and non-matched ESTs) were clustered using the SeqMan Pro application of the DNASTAR Lasergene software (DNASTAR, Inc., Madison, WI, USA) using the classic assemble method with default parameters [44]. The toxin-like precursors were grouped into 11 families (family A–K) based on their sequence homology and cysteine frameworks using ClustalX [45], with the default alignment parameters.

### 4.4. Dorsal Root Ganglion (DRG) Neuron Preparation

DRG neuron preparation was performed as described in the previous study [46]. Briefly, Sprague Dawley (SD) rats of either sex weighing 120–150 g (Hunan SJA Laboratory Animal Co., Ltd., Changsha, China) were anesthetized by isoflurane and sacrificed by decapitation, the spine was exposed and cut along the vertebral foramen, then the spinal cord was removed and the DRGs near the intervertebral foramens at all levels were collected. After trimming the attached vessel, nerves roots and membrane, the DRGs were put into 3 mL Dulbecco’s Modified Eagle Medium (DMEM) (Invitrogen; Thermo Fisher Scientific, Inc., Waltham, MA, USA) containing 3.125 mg/mL Neutral Protease (Dispase; #LS02104, Worthington) and 5 mg/mL Collagenase Type 1 (#LS004194, Worthington), incubated at 37 °C with gentle shaking for 45–60 min. The DRG neurons were collected by centrifuging at 200× *g* for 5 min, resuspended with DMEM supplemented with 10% FBS (Fetal Bovine Serum), 1% PS (Penicillin-streptomycin) (all from Gibco; Thermo Fisher Scientific, Waltham, MA, USA), 30 ng/mL Nerve Growth Factor (NGF) (Sigma, Sigma-Aldrich, Saint Louis, MO, USA), and seeded onto poly-d-lysine (PDL) (Sigma-Aldrich, Saint Louis, MO, USA) and laminin (Sigma-Aldrich, Saint Louis, MO, USA) coated coverslips. The neurons were allowed to attach to the coverslip after 30 min incubation, and then 1 mL medium was added to flood the cells. Approximately 16–24 h post seeding, the neurons were ready for patch-clamp experiments. Animals were used according to the guidelines of the National Institute of Health for Care and Use of Laboratory Animals. The experiments were approved by the Animal Care and Use committee of the College of Medicine, Hunan Normal University. 

### 4.5. RP-HPLC Purification of L. coelestis Venom and MALDI-TOF Analysis of the Eluted Fractions

*L. coelestis* venom was dissolved in ddH_2_O to a final concentration of 5 mg/mL, centrifuged and subjected to RP-HPLC (reverse phase high-performance liquid chromatography) purification in a Hanbon HPLC platform (Hanbon Sci. and Tech., Huai’an, China) equipped with a semipreparative C18 Column (10 mm × 250 mm, 5 µm; Welch Materials Inc, Shanghai, China) using a 55 min acetonitrile (ACN) gradient from 10% to 65% at a flow rate of 3 mL/min. An amount of 1 µL fraction sample and 0.5 µL CCA (α-Cyano-4-hydroxycinnamic acid; Sangon Biotech, Shanghai, China) (20 mg/mL, dissolved in 50% ACN supplemented with 0.1% TFA) was sequentially spotted onto 96-well target plate, air-dried and analyzed in an AB SCIEX 5800 MALDI-TOF (matrix-assisted laser desorption/ionization time of flight) mass spectrometer (AB SCIEX, Foster City, CA, USA). Mass spectra were acquired in a reflectron mode with the following settings: pulse width, 20 ms; vacuum degree, 4 × 10^−7^ torr; acceleration voltage, 25 kV. The mass range was set to 1000 to 10 kDa to identify most of the venom peptides (the matrix peaks were with MW (molecular weight) < 1000 Da and venom proteins are with MW > 10 kDa). 

### 4.6. Electrophysiology

Whole-cell patch-clamp recordings were performed in an EPC10 USB patch-clamp platform (HEKA Elektronik, Lambrecht, Germany). Pipettes were prepared from glass capillaries using the PC-10 puller (NARISHIGE, Tokoya, Japan). The pipette capacitance effect was minimized by filling only the pipette tip with the pipette solution. A high-resistance seal between the cell and the pipette tip was formed by gently applying a small negative pressure inside the pipette by a connected soft rubber tube, and the whole-cell configuration was formed by gently breaking the cell membrane attached to the pipette tip using the ZAP function of the amplifier (450 mV, 0.15 ms) when a small negative pressure inside the pipette was maintained. Cells with seal resistance less than 1 GΩ after break-in were discarded to minimize the effect of the leak current. The fast and slow capacitance effects were sequentially canceled using the C_f_ and C_m_ compensation function of the amplifier. To minimize the voltage error in the recording circuit, the serial resistance (R_s_) after the break-in was kept at less than 7 MΩ, and 80% R_s_ compensation with a speed value of 10 µs was used. To focus on NaV, CaV, and K_V_ channels expressed in the nociceptor neurons, only DRG neurons with C_m_ capacitance less than 45 pF (small- to medium-sized neurons) were selected for recording. The pipette solution for recording the voltage-gated potassium currents contains (in mM): 140 KCl, 2.5 MgCl_2_, 11 EGTA, and 10 HEPES (pH = 7.3, adjusted with KOH); and the corresponding bath solution contains (in mM): 140 NaCl, 5 KCl, 1 MgCl_2_, 2 CaCl_2_, 10 Glucose, and 10 HEPES (pH = 7.3, adjusted with NaOH). The pipette solution for recording the voltage-gated sodium currents contains (in mM): 140 CsF,1 EGTA, 10 NaCl, 10 HEPES (pH = 7.4, adjusted with CsOH); and the corresponding bath solution contains (in mM): 140 NaCl, 5 KCl, 2 CaCl_2_, 1 MgCl_2_·6H_2_O, 10 HEPES, and 10 Glucose (pH = 7.4, adjusted with NaOH). The pipette solution for recording the voltage-gated calcium currents contains (in mM): 120 CsMeSO_4_, 11 EGTA, 10 HEPES, 2 Mg-ATP (pH 7.4 with CsOH); and the corresponding bath solution contains (in mM): 105 CsCl, 40 TEA-Cl, 2 CaCl_2_, 1 MgCl_2_, 10 Glucose, and 10 HEPES (pH = 7.4, adjusted with CsOH). Venom was applied by perfusion as described in our previous study [23], and bath solution perfusion was used as the control. 

### 4.7. Data Analysis

Electrophysiological data were acquired using the PatchMaster software (HEKA Elektronik, Lambrecht, Germany), analyzed using IgoPro 6.10A (WaveMetrics Inc., Portland, OR, USA), Excel 2019 (Microsoft Corporation, Redmond, WA, USA), and GraphPad Prism 9 (GraphPad Software, La Jolla, CA, USA). Data were presented as MEAN ± SEM, while n was presented as the number of separate experimental cells. 

## Figures and Tables

**Figure 1 toxins-15-00143-f001:**
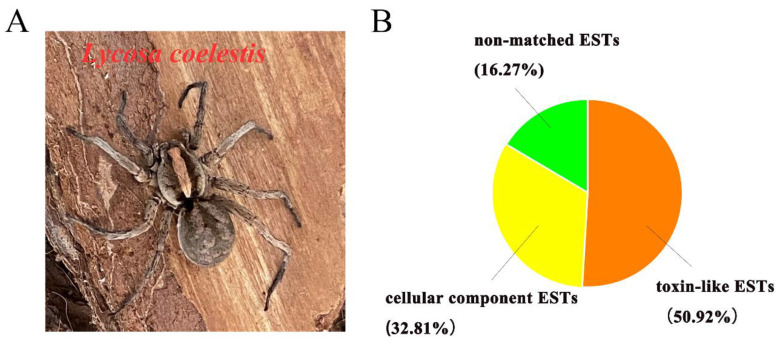
*L. coelestis* venom gland cDNA library ESTs distribution and annotation. (**A**) The spider *L. coelestis*. (**B**) The proportion of each category of ESTs derived from *L. coelestis* venom gland cDNA library sequencing.

**Figure 2 toxins-15-00143-f002:**
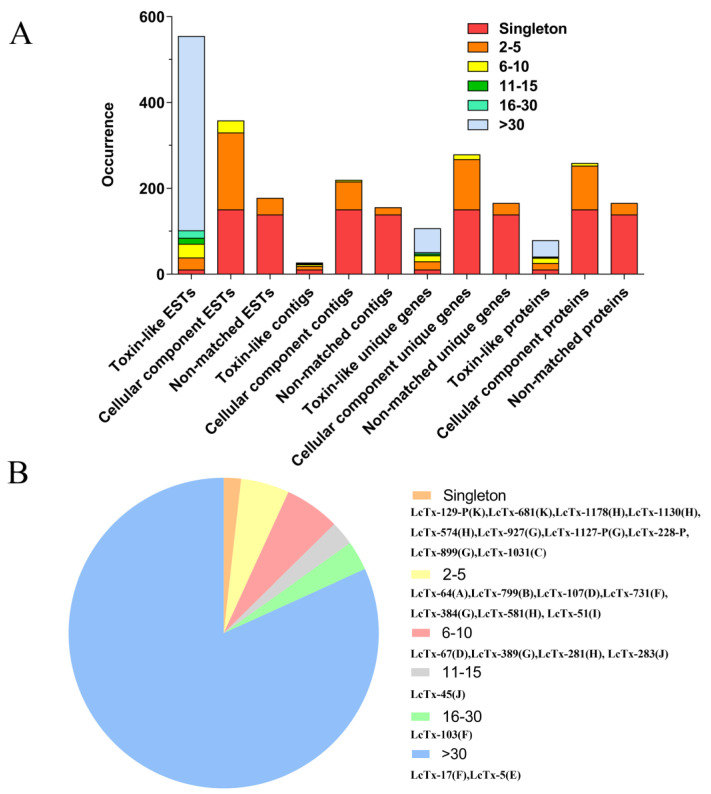
(**A**) Distribution of ESTs, contigs, unique genes, and proteins from the 3 functionally annotated categories (toxin-like, cellular component, and non-matched) among different cluster groups. (**B**) Representative toxin precursors in each cluster group; the letter in the bracket indicates the toxin’s family affiliation.

**Figure 3 toxins-15-00143-f003:**
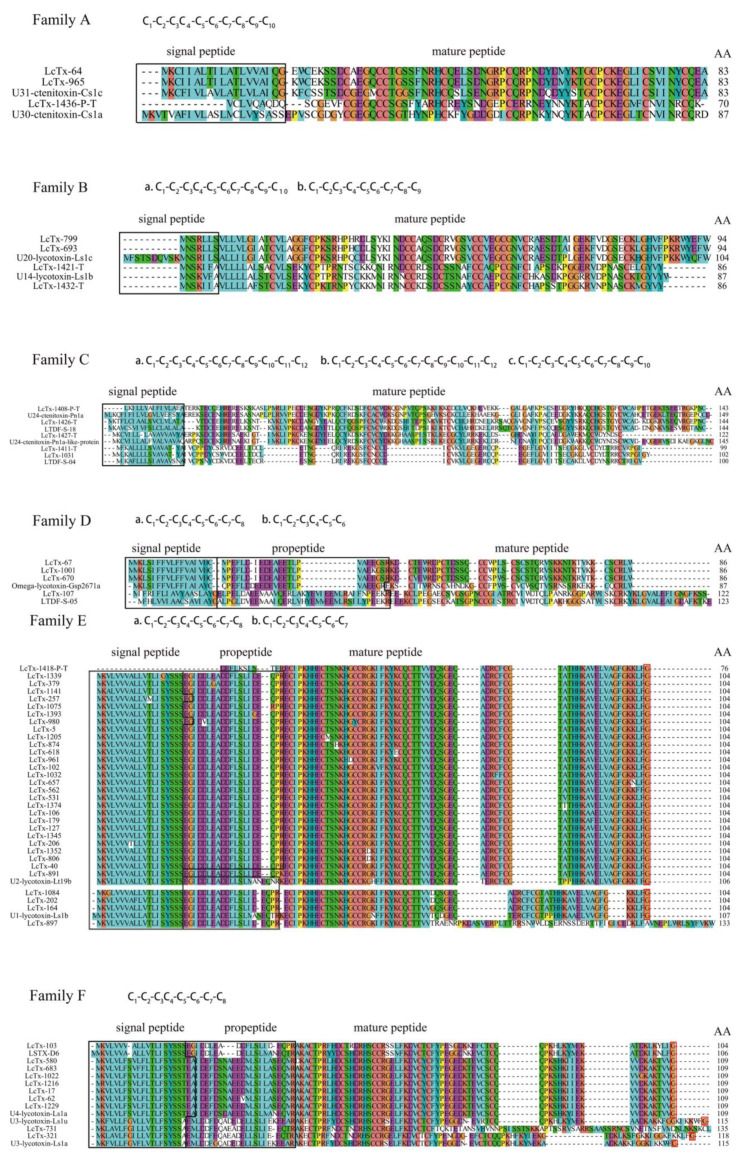
Family analysis of the toxin-like precursors derived from cDNA library and transcriptomic sequencing of the venom gland of *L. coelestis*. Consensus sequences in each family alignment are highlighted with colors, the black box region in each sequence indicates the signal peptide and/or propeptide, and the red box indicates the amidation signal (families E, F, H, I, J, and K).

**Figure 4 toxins-15-00143-f004:**
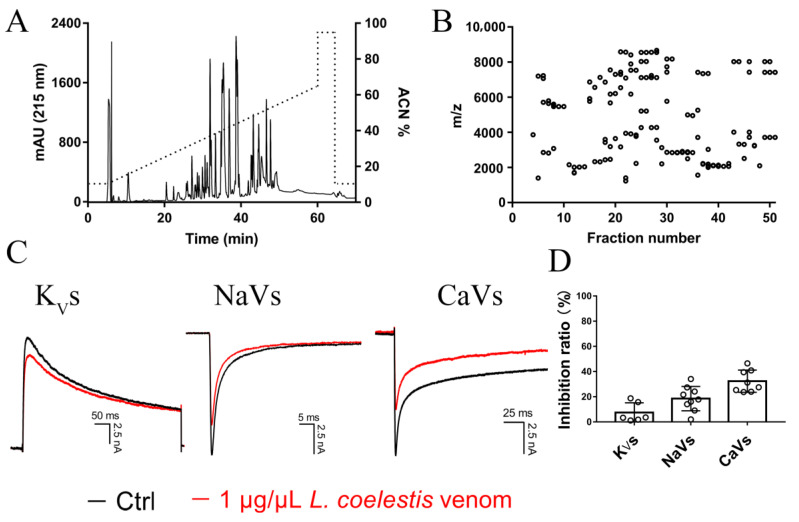
Venom peptides diversity in *L. coelestis* and the venom’s activity assay. (**A**) RP-HPLC profile of *L. coelestis* venom, dotted line indicates the acetonitrile (ACN) gradient. (**B**) The molecular weight distribution of the peptide toxins in RP-HPLC fractions from *L. coelestis* venom. (**C**) Representative current traces showing the effect of *L. coelestis* venom (1 μg/μL) on voltage-gated K^+^(K_V_), Na^+^(NaV), and Ca^2+^(CaV) channels in rat DRG neurons (n = 6–9); the black and red traces indicate the currents before (Ctrl) and after 1 µg/µL *L. coelestis* venom treatment, respectively. (**D**) Statistics showing 1 μg/μL *L. coelestis* venom inhibited the currents of DRG K_V_, NaV, and CaV channels by 7.45 ± 3.16%, 18.54 ± 3.23%, and 32.39 ± 3.12%*,* respectively (n = 6–9).

## Data Availability

The sequences of peptide toxins reported in this study were deposited in Genbank database with the accession numbers OQ207192—OQ207289.

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
