# Peer review of "The Molecular Composition of Peptide Toxins in the Venom of Spider Lycosa coelestis as Revealed by cDNA Library and Transcriptomic Sequencing"

_toxins, 2023, doi:10.3390/toxins15020143_

Round 1
Reviewer 1 Report
General comments
1. Use consistent fonts throughout the manuscript, even the links.
Results and discussions
1. Improve Figure 3 as it is not so visible and readable.
2. Improve and add more to the discussions in the sections 2.4 and 2.5
Materials and methods
1. Not even a single citation was provided in the section. Cite all the necessary citations in all the sections.
2. Methodology is not detailed and vague, like how much RNA was extracted. Do you directly assemble the short reads?? Usually sequences must be checked for adapters and other contaminants, and they must be removed.
3. No cut-offs were provided, and I am not able to see the accession ids of the submitted sequences. This should be made available to the public sequence database like NCBI. Provide necessary details.
4. How was clustering performed, and how many bootstraps?
5. What are DMEM and NGF?
6. Data analysis must include how many replicates and which statistical test was performed to compare the results.
Author Response
Reviewer#1:
Point 1. Use consistent fonts throughout the manuscript, even the links.
Response: Thanks, we have corrected it in this revision.
Results and discussions
Point 2. Improve Figure 3 as it is not so visible and readable.
Response: Thanks, we have revised Figure 3 (resolution increased to 3000 pixels/inch) to make it more clearer in this revision.
Point 3. Improve and add more to the discussions in the sections 2.4 and 2.5.
Response: Thanks for pointing it out.
We have added the discussion as “It’s hard to purify each of the peptide toxin in the venom to homogeneity even by combining multiple purification strategies (size exclusion chromatography, RP-HPLC, ion-exchange chromatography, and so on). Fortunately, the the peptide mass fingerprint as determined by MALDI-TOF MS analysis was commonly used for revealing the diversity of venom peptidomes and could be potential strategy for chemotaxonomy, with the count of the molecular species presented in the venom as revealed by MALDI-TOF analysis being an indicator of the molecular diversity of venom peptides (references: 1, DOI: 10.1016/j.jprot.2014.01.009; 2, https://doi.org/10.1016/j.euprot.2014.02.017; 3, DOI: 10.1002/jms.1389)” to section 2.4 in this revision (Lines 357-363).
For the experiments in section 2.5, we aimed at preliminarily uncovering the activity of Lycosa coelestis venom on ion channels endogenously expressed in primary cells, which we have applied to other venom studies in our previous studies (references: 1, DOI: 10.1016/j.toxicon.2016.11.252; 2, DOI: 10.3390/toxins6030988; 3, DOI: 10.1016/j.toxicon.2013.01.014; 4, DOI: 10.1016/j.toxicon.2014.06.001). We have added the discussion as “Spider venoms are rich in peptide toxins with potential modulatory activities on various ion channels, we have tested the activities of several spider venoms on endogenous ion channels expressed in primary cells in our previous studies(references: 1, DOI: 10.1016/j.toxicon.2016.11.252; 2, DOI: 10.3390/toxins6030988; 3, DOI: 10.1016/j.toxicon.2013.01.014; 4, DOI: 10.1016/j.toxicon.2014.06.001)” to this section (Lines 380-382).
Materials and methods
Point 4. Not even a single citation was provided in the section. Cite all the necessary citations in all the sections.
Response: Thanks, we have added appropriate citations in this section, please find them in Lines 423, 442, 443, 446, and 511.
Point 5. Methodology is not detailed and vague, like how much RNA was extracted. Do you directly assemble the short reads?? Usually sequences must be checked for adapters and other contaminants, and they must be removed.
Response: Thanks for pointing it out. The total extracted venom gland RNA is approximately 6.12 µg [A260/280 = 2.15, A260/230 = 1.79, RIN (RNA integrity number) is 9.4]. Adapters and low-quality reads were removed before performing the reads assembling. These details were added in this revision on Page 12, Lines 419-420, and 422.
Point 6. No cut-offs were provided, and I am not able to see the accession ids of the submitted sequences. This should be made available to the public sequence database like NCBI. Provide necessary details.
Response: Thanks for pointing it out. The nucleotide sequences of the toxin-like peptides were deposited in the Genbank database (https://www.ncbi.nlm.nih.gov/genbank/), and the assigned accession numbers are: OQ207192-OQ207289. These details were added in this revision on Page 5, Lines 166-169.
Point 7. How was clustering performed, and how many bootstraps?
Response: Thanks for pointing it out. The clustering analysis of ESTs by SeqMan Pro application of the DNASTAR Lasergene software and toxin-like peptides by Cluster X were performed using default parameters, these details were added in this revision on Pages 12-13, Lines 439-444.
Point 8. What are DMEM and NGF?
Response: Sorry we did not make it clear. DMEM and NGF are Dulbecco’s Modified Eagle Medium and Nerve Growth Factor, respectively. We have added these details on Page 13, Lines 452 and 458, in this revision.
Point 9. Data analysis must include how many replicates and which statistical test was performed to compare the results.
Response: Thanks. In the electrophysiology experiments in Fig. 4C-D, the number of separate experimental cell (n value) was indicated in the corresponding figure legend (Lines 375-377). We compared the effect of venom perfusion on the whole-cell currents (I) of NaV, KV, and CaV channels with that of bath solution perfusion (control), and the inhibition ratio (1-Ivenom/Ictrl) was summarized in Fig. 4D. We have added the text “Venom was applied by perfusion as described in our previous study, and bath solution perfusion was used as the control” in this revision, on Page 14, Lines 510-511.
Reviewer 2 Report
The manuscript reports interesting information about the composition of peptide toxins in the venom of spider Lycosa coelestis as well as on its biological activity as inhibitor of voltage-gated calcium channels. In general, it is well structured and written, and it represents a significant contribution at the basis of further discovery of active peptides, including voltage-gated calcium channels antagonists as potential analgesic agents.
Author Response
Reviewer#2:
The manuscript reports interesting information about the composition of peptide toxins in the venom of spider Lycosa coelestis as well as on its biological activity as inhibitor of voltage-gated calcium channels. In general, it is well structured and written, and it represents a significant contribution at the basis of further discovery of active peptides, including voltage-gated calcium channels antagonists as potential analgesic agents.
Response: Thanks for your kind review of our manuscript and these valuable comments.
Reviewer 3 Report
Manuscript "The molecular composition of peptide toxins in the venom of spider Lycosa coelestis as revealed by cDNA library and transcriptomic sequencing" is dedicated to to complex venomics study of the particular wolf spider naturally distributed in China, Korea and Japan. Authors utilize transcriptomics, HPLC and electrophisiology to annotate toxins structures and infer their effects on ion channels of DRG neurons. The results, reported in the manuscript can facilitate finding of new perspective biologically active peptides and proteins to be used as molecular instruments in biomedical and pesticide research.
There are several minor points to highlight:
1. Should the venom collection follow some formal agreement from ethics committee (section 3.1)?
2. Lines 36-37 I suggest to replace "globulin-like" to "globular" because globulins are members the particular protein family with the distinct common structure. None of the identified toxins seems to fit this protein family template;
3. Line 59 I did not understand what "talent strategies" means in such a context. Could this rather be "sophisticated"?
4. Line 63 The first proposition seems logically false to me. Spiders need paralyzing venom not because they predate on insects, but because they digest prey externally. Enzymes take time to digest substrate and toxins are the instruments that give spiders this time;
5. Line 72 Edman in "EDMAN degradation" is the surname of the researcher who inveted the technique, to the abbreviation;
6. Line 352 -353 Gradient parameters in text and in Fig. 4A don't match;
7. Line 389 Is "oebiotech" correct name of the firm?
Apart from these minor points article is mature enough to be published in "Toxins".
Author Response
Reviewer#3:
Manuscript "The molecular composition of peptide toxins in the venom of spider Lycosa coelestis as revealed by cDNA library and transcriptomic sequencing" is dedicated to to complex venomics study of the particular wolf spider naturally distributed in China, Korea and Japan. Authors utilize transcriptomics, HPLC and electrophisiology to annotate toxins structures and infer their effects on ion channels of DRG neurons. The results, reported in the manuscript can facilitate finding of new perspective biologically active peptides and proteins to be used as molecular instruments in biomedical and pesticide research.
There are several minor points to highlight:
Point 1. Should the venom collection follow some formal agreement from ethics committee (section 3.1)?
Response: Thanks. There is no mandatory regulation for the using and venom collection of the spider Lycosa coelestis and it’s not an endangered species. However, we only captured some of the mature spiders in the population in a given location to minimize the effect on environment.
Point 2. Lines 36-37 I suggest to replace "globulin-like" to "globular" because globulins are members the particular protein family with the distinct common structure. None of the identified toxins seems to fit this protein family template;
Response: Thanks for pointing it out, we have corrected it in this revision (Page 1, Line 41).
Point 3. Line 59 I did not understand what "talent strategies" means in such a context. Could this rather be "sophisticated"?
Response: Sorry we did not make it clear, we have revised it as “smart strategies” on Page 2, Line 64, in this revision.
Point 4. Line 63 The first proposition seems logically false to me. Spiders need paralyzing venom not because they predate on insects, but because they digest prey externally. Enzymes take time to digest substrate and toxins are the instruments that give spiders this time;
Response: Sorry for the ambiguity caused by our unclear statement. We have revised the text as “Spiders are one of the most successful arthropod predators. Their venom has been proven to be a rich source of hyperstable insecticidal mini-proteins that cause insect paralysis or lethality through the modulation of ion channels, receptors and enzymes” in this revision (Page 2, Lines 68-70).
Point 5. Line 72 Edman in "EDMAN degradation" is the surname of the researcher who inveted the technique, to the abbreviation;
Response: Thanks for pointing it out. We have corrected it on Page 2, Lines 77 and 79, in this revision.
Point 6. Line 352 -353 Gradient parameters in text and in Fig. 4A don't match;
Response: Thanks for pointing it out. We have corrected it as “as shown in Fig. 4A, 51 fractions were collected and most of them are with a retention time of 21-50 min (acetonitrile gradient from 26% to 55%)” on Page 11, Lines 366-369, in this revision. We also added the text “dotted line indicates the acetonitrile (ACN) gradient” in the legend of Fig. 4A (Page 11, Line 372).
Point 7. Line 389 Is "oebiotech" correct name of the firm?
Response: Yes, “oebiotech” is the correct name of the company.
Apart from these minor points article is mature enough to be published in "Toxins".
Round 2
Reviewer 1 Report
The manuscript can be accepted